# Effects of Respiratory Muscle Training on Baroreflex Sensitivity, Respiratory Function, and Serum Oxidative Stress in Acute Cervical Spinal Cord Injury

**DOI:** 10.3390/jpm11050377

**Published:** 2021-05-05

**Authors:** Hung-Chen Wang, Yu-Tsai Lin, Chih-Cheng Huang, Meng-Chih Lin, Mei-Yun Liaw, Cheng-Hsien Lu

**Affiliations:** 1Departments of Neurosurgery, Kaohsiung Chang Gung Memorial Hospital and Chang Gung University College of Medicine, Kaohsiung 833, Taiwan; m82whc@yahoo.com.tw; 2Otolaryngology, Kaohsiung Chang Gung Memorial Hospital and Chang Gung University College of Medicine, Kaohsiung 833, Taiwan; whc8131977@gmail.com; 3Graduate Institute of Clinical Medical Sciences, College of Medicine, Chang Gung University, Taoyuan City 33302, Taiwan; 4Neurology, Kaohsiung Chang Gung Memorial Hospital and Chang Gung University College of Medicine, Kaohsiung 833, Taiwan; hjc2828@gmail.com; 5Respiratory Therapy, Kaohsiung Chang Gung Memorial Hospital and Chang Gung University College of Medicine, Kaohsiung 833, Taiwan; linmengchih@hotmail.com; 6Division of Pulmonary and Critical Care Medicine, Department of Internal Medicine, Kaohsiung Chang Gung Memorial Hospital and Chang Gung University College of Medicine, Kaohsiung 833, Taiwan; 7Departments of Respiratory Care, Kaohsiung Chang Gung Memorial Hospital and Chang Gung University College of Medicine, Kaohsiung 833, Taiwan; 8Rehabilitation, Kaohsiung Chang Gung Memorial Hospital and Chang Gung University College of Medicine, Kaohsiung 833, Taiwan; 9Department of Biological Science, National Sun Yat-Sen University, Kaohsiung 833, Taiwan; 10Department of Neurology, Xiamen Chang Gung Memorial Hospital, Xiamen 361028, China

**Keywords:** cervical spinal cord injury, respiratory function, cardiovascular autonomic function, thiobarbituric acid-reactive substances

## Abstract

Background: respiratory complications are a leading cause of morbidity and mortality in individuals with spinal cord injury (SCI). We examined the effects of respiratory muscle training (RMT) in patients with acute cervical SCI. Methods: this prospective trial enrolled 44 adults with acute cervical SCI, of which twenty received RMT and twenty-four did not receive RMT. Respiratory function, cardiovascular autonomic function, and reactive oxidative species (ROS) were compared. The experimental group received 40-min high-intensity home-based RMT 7 days per week for 10 weeks. The control group received a sham intervention for a similar period. The primary outcomes were the effects of RMT on pulmonary and cardiovascular autonomic function, and ROS production in individuals with acute cervical SCI. Results: significant differences between the two groups in cardiovascular autonomic function and the heart rate response to deep breathing (*p* = 0.017) were found at the 6-month follow-up. After RMT, the maximal inspiratory pressure (*p* = 0.042) and thiobarbituric acid-reactive substances (TBARS) (*p* = 0.006) improved significantly, while there was no significant difference in the maximal expiratory pressure. Significant differences between the two groups in tidal volume (*p* = 0.005) and the rapid shallow breathing index (*p* = 0.031) were found at 6 months. Notably, the SF-36 (both the physical (PCS) and mental (MCS) component summaries) in the RMT group had decreased significantly at the 6-month follow-up, whereas the clinical scores did not differ significantly (*p* = 0.333) after RMT therapy. Conclusions: High-intensity home-based RMT can improve pulmonary function and endurance and reduce breathing difficulties in patients with respiratory muscle weakness after injury. It is recommended for rehabilitation after spinal cord injury.

## 1. Introduction

Pulmonary complications and cardiovascular autonomic function impairment are the most common acute systemic adverse events following spinal cord injury (SCI) [1,2]. They contribute to morbidity [3], mortality [3,4], and increased hospital stay length [1,2,4,5,6]. These changes are due to neuromuscular weakness [7] and may lead to lung infections [8]. Every 1% reduction in function increases the risk of death by 3% [1]. Respiratory adaptations appear to be largely accomplished within 3 weeks after injury [9].

Many studies have addressed respiratory challenges and management during acute SCI [10,11,12]. The respiratory manifestations of cervical SCI mainly depend on the level of injury [4,13,14], as manifested by reductions in the maximal expiratory pressure (MEP) and maximal inspiratory pressure (MIP), consistent with a restrictive ventilatory defect [15].

Inspiratory–expiratory pressure threshold respiratory muscle training (RMT) is a promising technique that positively affects the respiratory and cardiovascular dysregulation observed in people with chronic SCI [16]. Recent studies and meta-analyses have demonstrated that RMT can significantly improve respiratory muscle strength, function, and endurance during a training period [16,17,18,19]. As with able-bodied individuals, there is strong evidence supporting RMT for improved cardiovascular health in people with SCI [20]. Therefore, RMT may reduce the risk of developing SCI-induced lung disease by improving the ability to overcome airway restriction and obstruction and increasing respiratory endurance.

Evidence of oxygen free radical formation in animal SCI models demonstrates that reactive oxidative species (ROS) production increases shortly after SCI [21,22,23]. ROS appear shortly after SCI injury in rats, which is a critical time in the secondary pathophysiology of oxidative stress.

This study evaluated the effects of RMT on pulmonary and cardiovascular autonomic function and ROS production in adults with acute cervical SCI (first end point). We also evaluated the effects of RMT on functional outcomes using the Japanese Orthopedic Association (JOA) score recovery rate (second end point).

## 2. Materials and Methods

### 2.1. Patients

This is a prospective study. The diagnosis of acute cervical SCI is based on clinical symptoms and spine imaging, including X-ray, computed tomography (CT), and magnetic resonance imaging (MRI). Patients were excluded if they met the following conditions: (1) the time between injury and admission had exceeded 24 h; (2) had severe multiple trauma and unstable hemodynamic status; (3) had cervical trauma; (4) younger than 20 years old or older than 70 years old; or (5) had serious underlying diseases. Thirty patients were excluded from this study, including two with previous cervical trauma, three with severe multiple trauma and unstable hemodynamic status, four with severe underlying disease (3 with end-stage renal failure and one case with severe cirrhosis), ten of them were admitted to the hospital more than 24 h after the injury, and eleven refused to participate.

### 2.2. Ethical Approval and Consent to Participate

A total of 44 adult patients were included in the study. Injury mechanisms included 23 traffic accidents, 14 fall accidents, and three collisions with heavy objects. The Ethics Committee of the hospital’s Institutional Review Board approved the study (103-5218B and 106-1104C).

### 2.3. Assessment of Respiratory Muscle and Lung Function

During the assessment of respiratory muscle and lung damage, we monitored the following values: (1) MIP: MIP is measured with a pressure gauge after maximal expiration near residual volume. Respiratory therapists usually evaluate three efforts [24,25]. An MIP of 0 to −20 cmH2O is considered insufficient to produce the tidal volume (TV) needed to produce a good cough. (2) MEP: using the same technique as MIP measurement, the patient is instructed to exhale forcedly at full lung inflation (when the lung is totally inflated), and this is usually measured three times [26,27]. An MEP less than 30 cmH2O is considered inadequate for creating a good cough. (3) RSBI: the respiratory therapist uses Haloscale for the measurement. Calculated by dividing the breathing frequency by the TV [28,29].

### 2.4. Respiratory Muscle Training Protocol 

Respiratory muscle training was performed using Dofin Respiratory Trainer equipment (GaleMed Corporation, Taipei, Taiwan) [30]. The breathing device allows patients to train inspiratory and expiratory muscles and adjust the load independently, and provides a workload of up to 70 cmH2O by adjusting the load to 10 cmH2O [31,32].

The experimental group performed 40-min high-intensity home respiratory muscle training every day for 10 weeks. It can provide a workload of up to 79 cmH2O during inspiration and up to 82 cmH2O during expiration (reference: The DT11 has a pressure range of 5–39 cmH2O during inspiration and 4–33 cmH2O during expiration, while DT14 has a pressure range of 5–79 cmH2O during inspiration and 4–82 cmH2O during expiration). The experimental group received home-based RMT for a total of 40 min per day. The 40-min daily training was divided into two daily courses (i.e., morning and afternoon) and tailored for each participant. The daily session comprised 4 four-minute sets of respiratory training with one-minute rest intervals between each set. For inspiratory and expiratory intensity training, the initial training load was set to 50% of the baseline maximal inspiratory and expiratory pressure of each participant. Once a week, during a visit from a physiotherapist, the clinician measured the new values of the maximum inspiratory and expiratory pressure, and adjusted the load to 50% of the new value. The participants were blinded to the training load. To assess the strength of the respiratory muscles, the maximum respiratory pressure was measured using an analog vacuum manometer (Criticalmed, Rio de Janeiro, Brazil). All subjects performed 3 technically acceptable and repeatable tasks. For data analysis, the maximum value is used [18]. The control group undertook a sham RMT program with equivalent duration and scheduling of training. The sham group performed 40-min respiratory training every day for 10 weeks using the same Dofin Respiratory Trainer equipment (GaleMed Corporation, Taipei, Taiwan), but with no pressure load.

Outcomes were collected by a researcher who was blinded to the group allocation at baseline (two weeks after injury), after intervention (one month and three months after injury), and beyond intervention (six months after injury). Modified Borg scales were used to evaluate dyspnea severity during activity, graded from 0 (absence of dyspnea during strenuous exercise) to 10 (dyspnea during daily activities) [33,34,35].

### 2.5. Assessment of Autonomic Function

All patients with acute cervical SCI underwent a standardized evaluation of cardiovascular autonomic function, as described by Low [36]. All patients were tested between 9:00 a.m. and 12:00 p.m. Coffee, food, alcohol, and nicotine intake was forbidden for 4 h before the test. The test includes heart rate response to deep breathing (HRDB), Valsalva maneuver (VM), and 5-min resting blood pressure and heart rate recording for frequency domain analysis and spontaneous baroreflex sensitivity (BRS). Patients who are required to take a drug that is known to cause orthostatic hypotension or otherwise affect autonomous examinations had to stop taking the drug for five half-lives before the test, but only this was not harmful to the patient’s health.

Heart rate was derived from a continuously recorded standard three-lead electrocardiogram (ECGs) (Ivy Biomedical, model 3000; Branford, CT, USA), while continuously measuring arterial blood pressure (BP) using beat-to-beat photoplethysmographic recordings (Finameter Pro, Ohmeda; Englewood, OH, USA). The tests calculated by Testworks (WR Medical Electronics Company, Stillwater, MN, USA) obtained the following parameters—HRDB, Valsalva ratio (VR), and BRS obtained by Valsalva manipulation (BRSVM) [36]. BRSVM was derived from heart rate and blood pressure changes in the early stage II of VM by applying least-squares regression analysis.

Baroreflex sensitivity analysis software (Nevrokard, Slovenia) was used to spontaneously calculate the BRS using the sequence method (BRSseq) [37]. The program sets the following criteria when computing BRSseq: (1) systolic blood pressure (SBP) change greater than 1 mmHg, (2) sequence greater than 3 beats, and (3) correlation coefficient greater than 0.85. Both bradycardia (an increase in SBP that causes an increase in the R-R interval (RRI)) and tachycardia (a decrease in SBP that causes a decrease in RRI) sequences that fulfilled the criteria were enrolled. RRI and SBP fluctuations were synchronized in some subjects, but there is a time lag between these two fluctuations in other subjects. Therefore, for each subject, the BRSseq is calculated using a synchronization pattern and a shift pattern from 1 to 6 heartbeats [38]. The mode with the largest slope was selected. The average slope of the regression line was used as a measure of BRSseq.

The beat-to-beat RRI changes were interpolated using a third-order polynomial and resampled at 0.5-s intervals. A total of 512 samples were then used to transform the signal into the frequency domain by fast Fourier transform. The spectral power is divided into three frequency domains: high frequency (HF, 0.15–0.4 Hz), low frequency (LF, 0.04–0.15 Hz), and very low frequency (VLF, 0–0.04 Hz) [39]. The power ratio of LF and HF (LF/HF ratio) is regarded as an indicator of sympathetic vagal balance. All of the above measures were evaluated twice for the enrolled patients, before and after 3 months of RMT.

### 2.6. Blood Sampling and Assessment of Oxidative Stress

Blood samples were collected from 44 study patients within 24 h, 2 weeks, 1 month, 3 months, and 6 months after the onset of acute cervical SCI. The thiobarbituric acid-reactive substances (TBARS) uses a TBARS Assay Kit (Cayman Chemical, Ann Arbor, MI, USA) for rapid photometric detection at 532 nm of the thiobarbituric acid-malondialdehyde adduct. The measurement methods for oxidative stress have been described previously [40,41].

### 2.7. Clinical Manifestations and Quality-of-Life Assessment

The Japanese Orthopedic Association (JOA) cervical spine myelopathy function evaluation score was used to record the neurological condition at admission [42]. The Abbreviated Injury Score (AIS) for each body region was determined, and the total injury severity was calculated using the objective Injury Severity Score (ISS) at admission [43]. The choice of surgery depends on the imaging and clinical manifestations.

The questionnaire scale of this study was filled in by the patient or by the researcher after explaining the language, and then the Beck Depression Inventory (BDI) and SF-36 quality of life questionnaire were used [44,45].

### 2.8. Statistics

The category data were analyzed using the chi-square test or the Fisher exact test. Student’s *t*-test or the Mann–Whitney U test was used to analyze continuous data. Normally distributed data were mean ± standard derivation (SD) or median (inter-quartile range, IQR) for not-normally distributed data. Furthermore, a repeated-measures analysis of variance (two-way repeated measures ANOVA) was used to analyze the intra-individual course of parameters over time and to compare the parameters of two different groups (patients with or without RMT). To compare other continuous data between the RMT group and the non-RMT group at the time of admission and at the six-month follow-up, we used Student’s *t*-test. The SPSS 22.0 (SPSS Inc., Chicago, IL, USA) software was used for all statistical analyses.

## 3. Results

Table 1 summarizes the characteristics of the 44 adult acute cervical SCI patients. Most (36/44, 81.8%) were male, and their mean age was 45.4 (range 20–67) years. They were severely injured, as indicated by a median injury severity score (ISS) of 17 and a median JOA score of 5. The level of injury was from C2 to C7 in 4, 10, 14, 7, 6, and 3 cases, respectively. A total of thirty-eight patients underwent cervical spine surgery, including 23 emergency surgeries within 24 h after SCI and 32 elective surgeries. The mechanism of injury included 27 vehicle accidents, 13 falls, and 4 collisions. Of the patients, twenty (45.5%, 20/44) received 10 weeks of RMT. Between patients with and without RMT, no differences were found in age, sex, body mass index, SCI level, admission ISS, neurosurgical intervention, JOA scores at admission and 6 months, intensive care unit stay, SF-36 before discharge, Beck depression index before discharge and at 6 months, respiratory function before RMT, and serum TBARS at admission. However, significant differences were found between the two groups in hospitalization in the acute ward (*p* = 0.012), SF-36 at 6 months (*p* = 0.046 in PCS; *p* = 0.006 in MCS), and serum TBARS at 6 months (*p* = 0.024) (Table 1).

### 3.1. Baroreflex and Pulmonary Function

Table 2 shows the changes in cardiovascular autonomic function, pulmonary function, and functional scores during the study. No statistical difference in cardiovascular autonomic nerve function was found between the patients with and without RMT at baseline. However, significant changes were observed in the RMT group in the heart rate response to deep breathing (HRDB) (*p* ≤ 0.001), baroreflex sensitivity to the Valsalva maneuver (BRS–VM) (*p* = 0.047), in the low frequency/high frequency (LF/HF) ratio (*p* ≤ 0.001) after training, and HRDB (*p* = 0.017) at 6 months.

The respiratory function did not differ significantly between the two groups at baseline. However, the RMT group showed significant increases in MIP (*p* ≤ 0.001), MEP (*p* = 0.003), minute volume (*p* = 0.021), tidal volume (*p* = 0.033), and RSBI (*p* = 0.046) after training, and the non-RMT group showed significant increases in MIP (*p* ≤ 0.001) and MEP (*p* = 0.024) during the same period (Table 2). At 6 months, only tidal volume (*p* = 0.005) and RSBI (*p* = 0.031) differed significantly between the groups. A comparison of MIP and MEP showed a significant difference between the groups in MIP at 1 month (*p* = 0.010) (Figure 1A), but no significant difference in MEP at any time (Figure 1B).

### 3.2. TBARS

Examining the serum oxidative stress with TBARS revealed significant differences at 1 (*p* ≤ 0.001), 3 (*p* = 0.003), and 6 (*p* = 0.024) months between the two groups (Figure 2). Correlation analysis indicated that the serum TBARS levels at various time points were not significantly correlated with cardiovascular autonomic function, respiratory function, or the JOA recovery rate.

To compare the interaction of RMT and time in those dependent variables (pulmonary function, cardiovascular autonomic function, and biomarkers), we used two-way repeated measures ANOVA to analyze four time points (two weeks after injury, one month, three months, and six months). The results showed that time is a major independent factor and that RMT has a significant interaction with time in those dependent factors. RMT has a significant difference in MIP (*p* = 0.042) and TBARS (*p* = 0.006); however, there is no significant difference for RMT in other dependent factors.

## 4. Discussion

This study examined the changes in pulmonary and cardiovascular autonomic function and clinical scores in acute SCI patients before and after RMT. Several major findings are reported. First, cardiovascular autonomic function was significantly improved after RMT as measured by HRDB, BRS–VM, and the LF/HF ratio. However, at the 6-month follow-up, only HRDB differed significantly between the two groups. Second, after 1 month of RMT, MIP was significantly improved (Figure 1A), whereas MEP did not differ significantly between the groups at any time (Figure 1B). Third, respiratory function was significantly improved in both groups at the 6-month follow-up. The tidal volume (*p* = 0.005) and RSBI (*p* = 0.031) differed between the two groups at the 6-month follow-up. Fourth, at the 6-month follow-up, the SF-36 scores of the RMT group had decreased significantly, but the JOA recovery rate did not differ significantly between the two groups (*p* = 0.333). Furthermore, two-way repeated measures ANOVA showed that time is a major independent factor and RMT has a significant interaction with time in those dependent factors. RMT has a significant difference in MIP (*p* = 0.042) and TBARS (*p* = 0.006); however, there is no significant difference for RMT in other dependent factors.

### 4.1. Autonomic Function

The most important study finding was the improved cardiovascular autonomic function in patients with acute cervical SCI who received RMT. The HRDB, BRS–VM, and LF/HF ratio were significantly increased in the RMT group, although only the increment in HRDB differed significantly between the RMT and non-RMT groups (see Table 2). The improvement in HRDB may be attributed to the increased effort involved in deep breathing and the Valsalva maneuver, as the patients gained respiratory muscle strength. Recently, Legg Ditterline et al. showed that RMT improved both respiratory and cardiac autonomic function, ameliorating the cardiovascular stress response in those with chronic SCI [16]. They found significant increases in BRS in the RMT group without changes to the absolute peak or trough systolic blood pressure (SBP), which indicates a more effective baroreceptor response for the same change in SBP during the 5-s maximum expiratory pressure maneuver. This could potentially be a reversal of the vessel-stiffening common in SCI.

Significant decreases in the LF/HF ratio in our RMT group were consistent with the results reported by Legg Ditterline et al., who found significant increases in LH and HF power during the last 5 min of orthostatic stress in the RMT group (from 284 to 432 in LH and from 169 to 333 in HF), and the LF/HF ratio decreased from 1.68 to 1.29 after RMT. In another study, orthostatic hypotension was alleviated during the last 5 min in the seated position [46]. Those authors speculated that the increased HF power was the result of increased cardiovagal reflex activity and that the LF oscillation was caused solely by baroreceptor activity, as sympathetic recruitment by the baroreceptors increases sigmoidally as SBP drops [16,47]. Another study showed that reduced baroreflex sensitivity is closely related to increased arterial stiffness in patients with SCI [48]. Oscillation of the R–R interval in the LF band is attributed to sympathetic nervous system activity and possibly to modulation of the heart rate via the activity of the baroreceptor reflex loop [49,50,51]. Similar to our data, which showed increased sympathetic control over the heart (increased BRS–VM after RMT); Legg Ditterline et al. also detected significant increases in sympathetic control of the heart rate. The difference between that study and ours was that no orthostatic stress test was employed in our study. Another study detected baroreflex dysfunction in SCI patients only during orthostatic challenge [52]. While our patients started RMT 2 weeks after their injuries, those in the Legg Ditterline et al. study had chronic SCI. The clinical significance of these findings leads us to believe that RMT can reverse the baroreflex disorder caused by SCI. RMT appears to have more benefits in acute SCI compared to chronic SCI, but further analysis is needed.

### 4.2. Respiratory Function

Patients with acute cervical SCI have restrictive ventilatory impairment at admission [53,54,55]. One recent study found that expiratory muscle function was more compromised than inspiratory muscle function in those with tetraplegia and high paraplegia, and expiratory muscle training and electrical stimulation of the expiratory muscles improved the physiological parameters and cough function [55]. Phrenic nerve pacing or stimulation via intramuscular diaphragmatic electrodes can improve inspiratory muscle function [56,57,58]. However, these are invasive procedures and may cause phrenic nerve injury. Our study showed significant increases in tidal volume and RSBI at 6 months after training in the RMT group. As stated, respiratory muscle training started two weeks after injury. Comparing MEP improvement from two weeks to six months, there was slightly greater improvement in the RMT group (mean 60.9 cmH2O at two weeks and 92.6 cmH2O at six months, improved 31.7 cmH2O) than in the non-RMT group (mean 69.2 cmH2O at two weeks and 94.1 cmH2O at six months, improved 24.9 cmH2O). However, MEP is not significantly improved with the treatment. This is probably because the patients in the RMT group started at a lower level of MEP than the non-RMT group or because the number of patients is too small to reach statistical significance. Further large-scale studies may be warranted. Two-way repeated measures ANOVA showed that time is a major independent factor in MIP. RMT also has a significant difference in MIP and has a significant interaction with time. Furthermore, compared with the non-RMT group, the RMT group showed significant improvements in MIP at 1 month and in tidal volume and RSBI at 6 months. Several recent studies showed that RMT can significantly improve respiratory function based on MIP, MEP, and FVC with 4–12 weeks training [54,59,60]. In 67 individuals with SCI at levels C4–T12, Raab et al. found that at least 10 individualized inspiratory muscle training sessions produced a 7% increase in MIP and a 6.8% increase in MEP per 10 units of increase in training intensity [59]. Shin et al. admitted 104 patients with SCI for 4–8 weeks of in-patient clinical rehabilitation [60]. At follow-up, the percentage predicted value of FVC in the supine and sitting positions increased by 11.7% and 12.7%, respectively, on average. The peak cough flow improved by 22.7%. All assessed respiratory function parameters improved significantly in all subgroups [60].

One meaningful finding of our study was that MIP improved significantly in patients after 2 weeks of RMT. Due to the decrease in inspiratory muscle strength, the forced expiratory volume in 1 s (FEV_1_) and forced vital capacity (FVC) of SCI patients with higher-level lesions are reduced [13,14], and the reduction is correlated with the injury level [13]. Smoking and longer duration of injury are associated with greater reductions in FEV_1_ and FVC [14]. One study showed that a greater MIP is associated with higher FEV_1_ and FVC and is an important determinant of lung function [7]. Other researchers found that 14–50% of patients developed atelectasis or pneumonia during the acute period after SCI and had prolonged hospital stays [5,8,61]. Compared to the non-RMT group, the RMT group had the greatest increase in inspiratory (28 cmH_2_O) strength at 2 weeks beyond training (see Figure 1A). Therefore, RMT may reduce the risk of SCI-induced lung disease by improving the ability to overcome airway obstruction and increase breathing endurance, as well as significantly shortening the hospital stay (12.1 vs. 20.5 days in the RMT and non-RMT groups, respectively). Another recent study showed that patients with chronic SCI improved their coughing ability, breathing, speech skills, and overall quality of life after breathing training [16]. Those patients receiving RMT had significantly improved pulmonary function and shorter hospital stays.

### 4.3. Injury Severity and Other Co-Mobility

No significant differences in baseline parameters (including age, injury level, ISS severity, JOA score, and TBARS level) at admission were found between patients who did and those who did not receive RMT. However, the hospital stay was significantly longer in the non-RMT group. It is not surprising that improved respiratory function and a reduction in pulmonary complications can shorten the length of hospital stay in the acute ward. Similar to previous studies [16,54,60], we found no significant differences in motor function outcome for patients who did or did not receive RMT. The level of injury was a significant predictor of recovery, and the recovery rate was positively correlated with longer follow-up duration. Studies with follow-ups of 6 months or less reported significantly lower recovery rates for incomplete SCI compared to studies with 3–5-year follow-ups [62]. Factors, such as pneumonia [61], duration of injury [14], weight, and health status [63,64] contribute significantly to pulmonary dysfunction, exacerbating functional impairment due to SCI. One study showed that pulmonary complications did not affect the long-term change in ASIA Impairment Scale grades [5]. Our study showed no significant difference in the JOA score between the two patient groups at admission or at the 6-moth follow-up. This may help explain the lack of a significant relationship between functional improvement (JOA recovery rate) at 6 months in the patients who did or did not receive RMT.

Disability after traumatic spinal cord injury results from physical trauma and from “secondary mechanisms of injury” such as low metabolic energy levels, oxidative damage, and lipid peroxidation. In the rat model of traumatic spinal cord injury, increased levels of thiobarbituric acid reactive substances (TBARS) are associated with spinal cord tissue destruction and functional defects [65]. Tissue levels of thiobarbituric acid reactive substances (TBARS) [66] were significantly increased in experimental spinal cord injury. TBARS in the RMT group improved significantly beginning 2 weeks after respiratory training, and the improvement continued until the 6-month follow-up. Similar results were found in a study by Kahraman et al., where methylprednisolone was not able to lower these levels; however, hyperbaric oxygen therapy diminished all measured parameters significantly when compared with sham animals [66]. Although previous studies showed that serum TBARS levels in central nervous system diseases, such as traumatic brain injury [67,68,69], spontaneous intracranial hemorrhage [70], ischemic stroke [71], and Parkinson’s disease [72], are associated with clinical outcome, there were no significant differences between TBARS and respiratory function (MIP and MEP) in the present study. TBARS levels were not correlated with ICU stay or the length of hospitalization. There was a significant difference in TBARS levels in patients who received RMT compared to those who did not receive RMT, and RMT has a significant interaction with time. According to the study by Kahraman et al., and our results, increased oxygen levels, respiratory muscle training, or hyperbaric oxygen therapy seems to promote the prevention of oxidative spinal cord injury. However, the underlying mechanisms need further study.

Notably, the SF-36 (both the physical (PCS) and mental (MCS) component summaries) in the RMT group had decreased significantly at the 6-month follow-up (Table 1). Though patients in the RMT group exerted more effort, compared to the non-RMT group, limb strength did not recover significantly. This may explain why SF-36 in the RMT group decreased at the 6-month follow-up. Therefore, more attention should be paid to the psychological state of patients during their rehabilitation.

### 4.4. Study Limitations

This study has some limitations. First, it was not a randomized blind control study. In this case, the two groups of patients did not differ significantly, either demographically or clinically. Second, the study examined only the short-term (12 weeks) effects of RMT. Finally, there is no real evidence of a causal relationship between RMT and reactive oxidative species. In addition, the true quality-of-life changes were not investigated carefully. Prospective multicenter studies with longer follow-ups are needed to understand how long the effects of RMT persist for.

## 5. Conclusions

High-intensity home-based RMT can improve pulmonary function and endurance and reduce breathing difficulties in SCI patients with respiratory muscle weakness. Time is a key factor in pulmonary function recovery, and RMT leads to significant improvement in pulmonary function and has a significant interaction with time. It can also improve cardiovascular autonomic functioning during the acute SCI stage. It is recommended for rehabilitation in acute spinal cord injury.

## Figures and Tables

**Figure 1 jpm-11-00377-f001:**
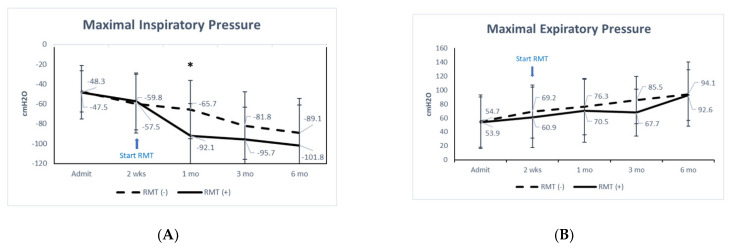
(**A**) Maximal inspiratory pressure and (**B**) maximal expiratory pressure (present with mean and standard derivation) at admission, two weeks, one month, three months, and six months in patients who received or did not receive RMT. * *p* < 0.05, by two-way repeated measures ANOVA.

**Figure 2 jpm-11-00377-f002:**
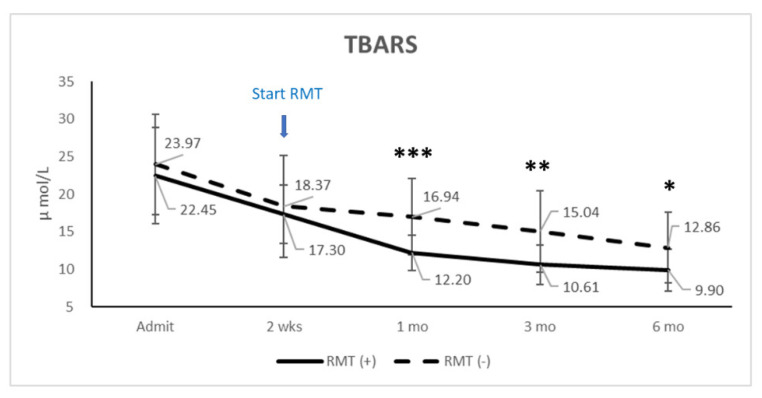
TBARS (present with mean and standard derivation) at admission, two weeks, one month, three months, and six months in patients who received or did not receive RMT. * *p* < 0.05; ** *p* ≤ 0.01; *** *p* ≤ 0.001, by two-way repeated measures ANOVA.

**Table 1 jpm-11-00377-t001:** Comparison between groups of patients with RMT and without RMT.

Parameters	RMT (N = 20)	No RMT (N = 24)	*p*-Value	Odds Ratio	95% CI (Lower, Upper)
Age (y)	46.1 ± 14.0	44.8 ± 15.5	0.788		−7.857, 10.291
Male	15	21	0.436	1.500	0.774, 2.906
Body mass index	25.8 ± 7.6	25.3 ± 4.8	0.762		−3.240, 4.391
Level of SCI			1.000	1.133	0.548, 2.343
C2-C4	15	19			
C5-C7	5	5			
Injury Severity Score at admission					
Total, Median (IQR)	20 (16, 21)	17 (16, 20)	0.196		
AIS-Head/Neck	4 (4, 4)	4 (4, 4)	0.944		
AIS-Thorax	0 (0, 0)	0 (0, 0)	0.563		
Neurosurgical intervention					
Total	18	20	0.673	0.704	0.216, 2.291
Emergent	12	11	0.382	0.730	0.373, 1.428
Elective	15	17	1.000	0.889	0.414, 1.909
JOA score at admission, Median (IQR)	5 (4, 6)	6 (4, 12)	0.539		
JOA recovery rate at 6 months (%)	53.2 ± 33.3	62.8 ± 29.9	0.333		−29.266, 10.154
ICU stay (days)	4.8 ± 3.3	5.6 ± 5.0	0.646		−3.960, 2.494
Hospitalization in acute department (days)	12.1 ± 5.3	20.5 ± 12.5	0.012		−15.130, −1.519
SF-36 before discharged					
PCS	17.6 ± 12.0	20.3 ± 19.4	0.233		−5.461, 13.857
MCS	39.6 ± 16.4	47.8 ± 12.2	0.113		−1.455, 11.033
SF-36 at 6 months					
PCS	13.5 ± 8.5	32.5 ± 25.8	0.046		−37.679, −0.447
MCS	32.4 ± 5.3	50.2 ± 16.2	0.006		−29.450, −6.121
BDI before discharged	27.6 ± 10.5	22.3 ± 9.2	0.112		−1.315, 12.065
BDI at 6 months	25.7 ± 12.2	19.6 ± 13.0	0.259		−4.761, 16.918
Respiratory function at time before RMT					
MIP (cmH2O)	−57.5 ± 28.8	−59.8 ± 38.1	0.355		−16.573, 21.263
MEP (cmH2O)	60.9 ± 38.1	69.2 ± 43.1	0.946		−34.624, 18.103
Respiratory rate (bpm)	15.3 ± 3.7	14.3 ± 4.6	0.885		−3.383, 6.379
Minute volume (L/min)	10.3 ± 3.1	10.6 ± 3.0	0.702		−5.404, 4.297
Tidal volume (L)	0.75 ± 0.22	0.80 ± 0.29	0.379		−0.424, 0.195
RSBI	31.7 ± 13.6	22.7 ± 14.4	0.752		−14.537, 25.990
TBARS at admission	22.45 ± 6.40	23.97 ± 6.67	0.459		−5.625, 2.586
TBARS at 6 months	9.90 ± 2.78	12.86 ± 4.72	0.024		−5.511, −0.404

**Table 2 jpm-11-00377-t002:** Changes in cardiovascular autonomic function and pulmonary function between groups of patients with RMT and without RMT during the study period.

	RMT Group (*n* = 20)		Non-RMT Group (*n* = 24)		p^￥^ between RMT and without RMT Groups at the Time of 6 Months Follow-Up
	Baseline	Follow-Up	Baseline	Follow-Up	
Cardiovascular autonomic function					
HR_DB	9.76 ± 2.07	13.09 ± 2.88 *	9.65 ± 3.85	10.30 ± 3.03	0.017
Valsalva ratio	1.54 ± 0.33	1.47 ± 0.30	1.54 ± 0.32	1.50 ± 0.29	0.794
BRS_VM	2.20 ± 1.50	2.77 ± 1.68*	2.16 ± 1.39	2.10 ± 1.48	0.274
BRS_Seq	11.39 ± 5.00	12.40 ± 6.87	10.73 ± 6.59	11.75 ± 7.20	0.810
LF/HF ratio	1.44 ± 0.57	1.15 ± 0.42*	1.50 ± 0.71	1.46 ± 0.62	0.132
Pulmonary function parameters					
MIP (cmH2O)	−57.5 ± 28.8	−101.8 ± 41.0 *	−59.8 ± 29.2	−89.1 ± 34.8 *	0.358
MEP (cmH2O)	60.9 ± 38.1	92.6 ± 45.9 *	69.2 ± 43.1	94.1 ± 36.3 *	0.925
Respiratory rate (bpm)	15.3 ± 3.7	15.8 ± 6.8	14.3 ± 4.6	13.2 ± 3.3	0.269
Minute volume (L/min)	10.3 ± 3.1	14.8 ± 5.5 *	10.6 ± 3.0	10.2 ± 4.9	0.073
Tidal volume (L)	0.75 ± 0.22	1.14 ± 0.35 *	0.80 ± 0.29	0.88 ± 0.31	0.005
RSBI	21.7 ± 13.6	12.7 ± 5.2 *	22.7 ± 14.4	32.7 ± 22.6	0.031
Borg scale					
Initial	1.33 ± 2.30	0.50 ± 0.71	1.08 ± 0.49	1.10 ± 0.55	0.109
Finish	2.67 ± 2.08	1.67 ± 1.53	3.50 ± 1.05	2.20 ± 1.30	0.616
Difference	1.33 ± 0.58	1.25 ± 1.26	2.41 ± 1.36	1.06 ± 1.07	0.779

Abbreviations: RMT, respiratory muscle training; SCI, spinal cord injury; CI, confidence interval; IQR, interquartile range; JOA, Japanese Orthopedic Association; MIP, maximal inspiratory pressure; MEP, maximal expiratory pressure; RSBI, rapid shallow breathing index; NA, not available. Data are presented as absolute numbers (mean ± SD). The changes (baseline and 6-month follow-up) of cardiovascular autonomic function and pulmonary function parameters in different groups (RMT and non-RMT) were compared using paired t-test, respectively. * Statistical significance (*p* < 0.05) between baseline and follow-up. p^￥^ Between RMT and without RMT groups at the time of 6 months follow-up.

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
