# Peer review of "Effects of Respiratory Muscle Training on Baroreflex Sensitivity, Respiratory Function, and Serum Oxidative Stress in Acute Cervical Spinal Cord Injury"

_jpm, 2021, doi:10.3390/jpm11050377_

Round 1
Reviewer 1 Report
My biggest concern would be the statistical analyses. The authors performed t-test or Mann-Whitney to compare between two groups. However, this is a longitudinal data. Thus, I disagree with statistical analyses of choice. Instead they should have chosen tests like two-way repeated measure ANOVA, to compare the rate of improvement between case vs control groups. The analyses needs to be controlled for confounders, such as use of steroid, ventilator dependence. Table 2 has baseline and follow up of RMT and non-RMT group. There is also a p-value. I am confused what does this p-value mean? Which comparison yield that p-value? MEP seems to be worse in RMT group, compared to non-RMT? How do you explain that? And that observation did not support your conclusion that stated "RMT can improve respiratory muscle strength"...
Reviewer 2 Report
Wang and colleagues presented an interesting study on the use of respiratory muscle training (RMT) on acute cervical spinal cord injury (SCI) patients. Authors demonstrated that RMT improved cardiovascular autonomic functions and has some positive effect on the respiratory function of SCI patients. This study will increase the body of literature that indicates that RMT is an effective rehabilitative training after spinal cord injury. The authors describe the study's limitations well. There are some issues on the manuscript that the authors should improve:
1) Please described the sham RTM program.
2) The RTM group had a significant less hospitalization days than the non-RTM patients. Authors should discuss how this could affect their results.
3) What day after the injury did the RTM (or sham) program start? It was roughly the same day post injury or there were significant differences?
4) Authors observed an interesting decrease on TBARS in patients who received RMT compared to the ones that did not received RMT. However, the differences were not correlated with improved in function. How did the RMT program decrease TBARS levels? It was not even clear why the authors went to study this process (the justification on the paragraph on the introduction section could be applied to many secondary SCI pathophysiological events). Did other studies observed the same on cervical spinal cord injury patients? The references provided on the discussion section are not from SCI. Are TBARS levels correlated with length of hospitalization? Authors should improve the discussion on this part.
5) Authors did not perform any quality of life assessment. Did the observed improvement have a significant impact on patients' quality of life?
Round 2
Reviewer 1 Report
N/A
Author Response
Reviewer 2
Comments and Suggestions for Authors: N/A.
Reviewer 2 Report
The authors accurately addressed my concerns. There is only one issue remaining. The results and conclusion of the SF-36 questionnaire (namely the MCS) should be mentioned on the abstract.
Author Response
Reviewer 2
Comments and Suggestions for Authors
The authors accurately addressed my concerns. There is only one issue remaining. The results and conclusion of the SF-36 questionnaire (namely the MCS) should be mentioned on the abstract.
Ans: Thanks for your comments. We added the results and conclusion of the SF-36 questionnaire (namely the MCS) on the abstract.